# Does low fertility indicate better reproductive health status? Evidence from nationally representative survey in India

Roni Sikdar👤*, Dhananjay W. Bansod👤

Department of Public Health & Mortality Studies, International Institute for Population Sciences, Mumbai, Maharashtra, India

* ronisikdar16@gmail.com

## Abstract

### Background

The global demographic landscape is experiencing a significant transformation of declining fertility rates, which has far-reaching implications for societal development and women's well-being. The study investigates the association between declining fertility rates and women's reproductive health in India, considering socioeconomic and demographic factors as well as regional variations.

### Methods

The study uses data from the recent National Family Health Survey (NFHS-5) round conducted during 2019−21. A composite index called the Reproductive Health Index (RHI) is constructed by equally weighing indicators such as antenatal care, anemia, and body mass index. To evaluate the robustness of this index, a sensitivity analysis is performed. Descriptive statistics and Poisson regression models are employed to explore the association between fertility and RHI among currently married women.

### Results

The findings show substantial differences in RHI scores across socio-economic, demographic groups, and geographical regions. The lowest RHI score of 4.09 is found in the Eastern region, whereas those in the Northern region exhibit the highest score of 4.42. The analysis further indicates a negative relationship between fertility and reproductive health. Women with four or more children exhibit an RHI score of 1.97 compared to 2.98 among those with one child. The Poisson regression analysis indicates that women with at least four children have lower RHI scores, even after adjusting for socio-economic and demographic factors. Women in wealthier quintiles and those with media exposure report significantly higher RHI score compared to those in the poorest wealth quintiles and no media exposure.

**Data availability statement:** The dataset used in this study are publicly available and can be accessed from DHS Program: https://dhsprogram.com/. Any researcher can access this dataset from DHS after registering at: https://dhsprogram.com/data/new-user-registration.cfm and get permission to access the data at: https://dhsprogram.com/data/dataset/India_Standard-DHS_2020.cfm?flag=1.

**Funding:** The author(s) received no specific funding for this work.

**Competing interests:** The authors have declared that no competing interests exist.

## Conclusion

In conclusion, this research highlights the critical need for targeted interventions to address regional and socio-economic inequities in healthcare access and reproductive health services. By exploring the intricate relationship between low fertility and reproductive health, this study contributes to the discourse on gender equality, reproductive rights, and sustainable societal development. The findings provide evidence to guide public health policies and programs designed to promote women's reproductive health.

## Introduction

The Sustainable Development Goals (SDGs) established by the United Nations emphasise the importance of gender equality and the empowerment of all women and girls. It highlights the need for equitable access to resources and services that enhance the overall well-being of all individuals by 2030. Within this framework, SDG-5 focuses on achieving healthy lives and promoting the well-being of all ages, especially women and girls. It highlights the need for universal health coverage and access to sexual and reproductive health care services [1]. Despite extensive research on fertility trends and their economic and demographic implications, the association between fertility transition and women's reproductive health remains underexplored.

Previous literature predominantly addresses the economic and societal aspects of fertility decline. Studies have shown that lower fertility rates can contribute to economic growth and development by enabling women to participate in the workforce [2–5]. However, these studies often overlook the nuanced health outcomes and personal experiences. For instance, while lower fertility is associated with improved maternal health and reduced mortality rates [6], how it affects women's overall well-being is less clear. This study addresses this gap by developing a composite Reproductive Health Index (RHI) to provide a comprehensive assessment of women's health in relation to fertility patterns.

The global demographic landscape is experiencing a significant transformation characterised by declining fertility rates [7–9]. Recently, India also experiences low fertility, as the latest National Family Health Survey (NFHS-5) shows that the total fertility rate (TFR) of India has decreased to 2.0 from 3.39 in 1992−93 (NFHS-1) [10,11]. The transition towards low fertility rates makes a sharp divergence from India's long-standing image of high fertility, restructuring the country's demographic composition [12]. The decline is indicative of broader socio-economic transformation, including increased access to education, improved healthcare services, and enhanced economic opportunities for women [13–20]. However, the implications of these changes on women's reproductive health are complex and multifaceted. This requires closer examination.

A unique aspect of this study is its focus on geographical variations in the impact of low fertility on women's well-being across different regions of India. Previous

studies largely focus on national averages, failing to capture regional disparities [21–24]. By highlighting these differences, this research provides valuable insights for region-specific policy interventions. It helps to understand how different regions within the country experience and respond to fertility changes, which is crucial for developing targeted strategies that address the unique needs of women in diverse contexts.

Several theoretical perspectives inform this study. Yukiko Asada developed a framework to measure health inequity, which aims to understand the complexities of moral and ethical dimensions within broader systemic disparities in access to healthcare, nutrition, and education [25]. Studies indicate that socioeconomic status, regional healthcare infrastructure, and social stratification play a major role in health outcomes [26–28]. For women in rural or marginalised communities, the potential benefits of improved reproductive healthcare associated with fertility decline are often overshadowed by barriers such as inadequate access to antenatal care and persistent nutritional deficits [29,30]. The capability approach developed by Sen (1999) provides another approach to understanding how fertility decline intersects with women's well-being. Fertility reduction is often associated with enhanced access to education, workforce participation, and quality healthcare [31–33]. However, existing studies reveal a critical inequity: While fertility is linked with better health outcomes, these benefits are unevenly distributed. Women from lower socio-economic groups are often excluded from these advantages due to systemic barriers, including limited access to quality healthcare and socio-cultural constraints [34,35]. This underscores the importance of framing reproductive health not solely as an outcome but as a capability that requires active nurturing through equitable and inclusive policy interventions [36,37]. The life course perspective further adds different sheds by exploring how early life events and cumulative disadvantages shape reproductive health trajectories. Early marriages, high fertility rates, and inadequate access to healthcare during reproductive stages significantly increases the risks of adverse health outcomes [38,39]. Women who experience these compounded disadvantages are less likely to get the benefit of low fertility, perpetuating reproductive health inequities. This framework highlights the importance of addressing structural disadvantages across all stages of life to ensure that the benefits of low fertility are equitably distributed [40,41].

The World Health Organisation defines reproductive health as a state of complete physical, mental, and social well-being and not merely the absence of disease or infirmity in all matters relating to the reproductive system [42]. This comprehensive definition underscores the necessity of considering a wide range of health indicators when assessing women's reproductive health. Despite extensive research on the socio-economic consequences of fertility decline, there remains limited evidence on its direct implications for women's reproductive health. Most of the studies focus on economic benefits, workforce participation, and maternal mortality but do not capture broader health indicators. Furthermore, national studies highlight state-level differences and often lack district-level insights. This study addresses this gap by developing the Reproductive Health Index (RHI). RHI incorporates various indicators to provide a comprehensive assessment of women's health in relation to fertility.

By integrating these various dimensions, this study aims to offer a more comprehensive understanding of the impact of low fertility on women's reproductive health. We set out to answer two research questions: (1) What is the extent of RHI in India? (2) How does low fertility impact the reproductive health of women in India, considering socioeconomic and demographic factors as well as regional variations? These findings underscore the need for targeted policies that address systemic inequities, promote reproductive rights, and ensure that the benefits of low fertility translate into substantive improvements in women's reproductive health.

## Methods

### Data sources

The study uses data from the NFHS-5 conducted in India in 2019−21. The International Institute for Population Sciences (IIPS) in Mumbai conducted this survey with funding from the Government of India's Ministry of Health and Family Welfare. The NFHS is a nationally representative household survey that provides comprehensive data for monitoring and

evaluating various indicators related to population, health, and nutrition. The survey design is optimized to ensure high precision in fertility and family planning indicators. The survey was approved by the Institutional Review Board (IRB) of the institutions involved. The detailed datasets are available at (https://dhsprogram.com/data/dataset/India_Standard-DHS_2020.cfm?flag=1). This datasets is entirely anonymized and does not contain any personally identifiable information about respondents, households, or communities. Therefore, ethical approval was not required for this study. The NFHS 2019−21 report was developed to offer assessments of essential indicators at the national level (28 states, 8 union territories) and district level (707 districts). Using a two-stage stratified sampling design, the NFHS −5 sample surveyed 636,699 households and 724,115 women of reproductive age with a 97% response rate [11].

In this study, women who report never having been in a union, divorced, or separated and have no children are excluded. Our analytic sample includes an average of 173,966 women aged 15–49 years who are currently married or in a union (hereafter referred to as partnered women) who gave birth to at least one child in the 5-year period preceding the survey.

## Study variables and their measurement

We use the women's module to explore the reproductive health status of women. This module includes questions about medical attention at birth, antenatal coverage, ever-experienced stillbirth, early neonatal mortality, the prevalence of anemia, body mass index (BMI), low birth weight, and delivery by cesarean section. In NFHS, BMI is calculated from measured height and weight, while anemia status is determined from haemoglobin levels, and birth outcomes based on reported maternal health experiences. Skilled or unskilled care during antenatal and delivery is obtained from provider-based questions, while cesarean delivery and history of miscarriage or stillbirth serve as additional reproductive health indicators. We exclude abortion from the miscarriage/stillbirth indicator to maintain focus on involuntary and health-related pregnancy losses. We consider these indicators (Table 1) to measure RHI [22–24,43,44]. The inclusion of these factors is essential for comprehensively assessing women's reproductive health status. We compute an individual-level score by adding 7 indicators from 7 questions in the NFHS-5. Equal weights are assigned to each variable, indicating that each is considered equally crucial for measuring women's reproductive health. Each indicator is associated with a specific threshold value, beyond which a woman is assigned an RHI score. However, it is essential to recognise that women's RHI is a complex and unfolding process characterised by a progression rather than a simple binary classification. Therefore, we anticipate that our findings will hold significance. To determine the weights for dichotomous variables, a particular approach is employed wherein women facing any adverse reproductive health outcome incur a penalty of 0, while those experiencing any positive reproductive health outcome receive a reward of 1.

The explanatory variable is the number of children ever born (CEB). Further, we include women's present age, age at first marriage, age at first birth, caste categories, religion, place of residence, and regions like North (Chandigarh, Delhi, Haryana, Himachal Pradesh, Jammu & Kashmir, Ladakh, Punjab, Rajasthan, and Uttarakhand), East (Bihar, Jharkhand, Odisha, and West Bengal), Northeast (Arunachal Pradesh, Assam, Manipur, Meghalaya, Mizoram, Nagaland, Sikkim, and Tripura), Central (Chhattisgarh, Madhya Pradesh, and Uttar Pradesh), West (Dadra & Nagar Haveli and Daman & Diu, Goa, Gujarat, and Maharashtra), and South (Andaman & Nicobar Islands, Andhra Pradesh, Karnataka, Kerala, Lakshadweep, Puducherry, Tamil Nadu, and Telangana) to effectively understand the unadjusted and adjusted association between low fertility and women's reproductive health.

## Statistical analysis

In the beginning, we illustrate the percentage distribution of each variable within the framework of this country. To ensure the findings accurately reflect the overall population, all results are weighted using the survey sample weights provided by the NFHS. Furthermore, we use a composite index to identify the principal contributors of RHI. This approach provides a comprehensive understanding of the critical determinants influencing women's reproductive health in the studied

**Table 1. Variables, Measurement, and Coding of Reproductive Health Index (RHI).**

| Variable | Definitions | Code or units |
|---|---|---|
| Medical attention at birth | It refers to a birth attended by different health personnel. It is essential to measure the diverse health needs of women accurately. | Yes = 1<br>No = 0 |
| Antenatal coverage | Antenatal coverage refers to the percentage of women who receive adequate care during pregnancy. It is important to evaluate women's access to essential health services, thereby providing a comprehensive measure of their reproductive health and well-being. | Yes = 1<br>No = 0 |
| Ever experienced stillbirth | It is essential to maternal health challenges and higfhlights disparities across different groups. Ever experience stillbirth refers to women who have reported experiencing a stillbirth at any point in their life. | Yes = 0<br>No = 1 |
| Prevalence of Anaemia in women | Anaemia refers to the proportion of women who have hemoglobin levels below the normal range. It is important to identify nutritional and health deficiencies. | Yes = 0<br>No = 1 |
| BMI | Body Mass Index is measured by height and weight to see the nutritional and health status across different groups. | Normal weight (18.5–24.9 kg/m2) = 1<br>Others (<18.5 kg/m2 & ≥ 25.0) = 0 |
| Low birth weight | It is defined as infants weighing less than 2.5 kg at birth, and it reflects maternal health and prenatal care quality, reflecting women's reproductive health. | Normal = 1<br>Below 2.5 kg = 0 |
| Delivered by cesarean section | Indicates medical intervention level and maternal healthcare quality, essential for assessing women's well-being. | Yes = 0<br>No = 1 |

Source: Adapted from (22–24,44). Note: This variable includes only stillbirth and miscarriage. Abortion is excluded to focus on involuntary pregnancy losses.

population (23,24,44). Additionally, we conducted bivariate analyses to assess the levels of women's reproductive health based on the socioeconomic and demographic characteristics of the mother.

$$RHI_I = \frac{1}{n}\sum_{j=1}^{n} X_{ij}$$

$RHI_I$ = Reproductive Health Index score for women $i$
$X_{ij}$ = Standarised value of reproductive health indicator $j$ for women $i$
$n$ = Total number of reproductive health indicators

In the next phase, understanding the count form of the interest variable, we utilise the Poisson regression model to examine whether respondents had access to distinct overall well-being.

$$\log(E(RHI_i)) = \beta_0 + \beta_1 CEB_i + \sum_{k=2}^{m}\beta_k Z_{ik} + \epsilon_i$$

$E(RHI_i)$ = Expected Reproductive Health Index for women $i$
$CEB_i$ = Number of children ever born (Key explanatory variable)
$Z_{ik}$ = Vector of socio-economic and demographic control variables
$\beta_k$ = Coefficients for the explanatory variables
$\epsilon_i$ = Error term

This analysis is divided into two main parts. First, we investigate whether having a lower CEB is associated with women's reproductive health. Second, we explore the association between women's reproductive health and the CEB while also considering background characteristics. To assess the robustness of the composite index, we conduct a sensitivity analysis by reconstructing the RHI using alternative indicator combinations (RHI2, RHI3, RHI4) and run the Poisson regression models. The results remain consistent and support the reliability of the index (S1 Table).

 

Before conducting the regression analysis, we tested for multicollinearity among key age-related covariate using the Variance Inflation Factor (VIF). All values are below the standard threshold, indicating that multicollinearity is not a concern. Additionally, we assess for overdispersion using the Pearson chi-square goodness-of-fit test. The dispersion value is close to 1 ($\chi^2 = 45,143.74$, $p = 1.000$), indicating no evidence of overdispersion.

We utilise Stata version 17.0 to conduct data analysis. We use ArcMap to create maps. The shapefile used to generate the map is freely available from the official Demographic Health Survey website (https://dhsprogram.com/data/available-datasets.cfm). The map is prepared by the author.

## Results

### Variation in individual RHI indicators within the country

Table 2 presents an overview of each indicator of women's well-being in India. Skilled providers, around 90.2 per cent, attend the majority of deliveries. However, 9.8 per cent of deliveries occur without skilled assistance, which highlights both healthcare gaps and the persistence of cultural or personal preferences for home delivery. Similarly, 85.1 per cent of women receive antenatal care from skilled providers, and 14.9 per cent receive no care or care from unskilled providers, indicating potential gaps in access to quality healthcare services. Furthermore, it highlights the significant maternal health challenge, with 74.6 per cent of women reporting experience of stillbirths or miscarriages within this domain (combined percentage of miscarriage, abortion, and stillbirth). This high prevalence underscores persistent challenges in maternal health and suggests the need for enhanced support and interventions. Anemia remains a significant concern despite

**Table 2. Weighted average percentages of each indicator for women's reproductive health in India (2019–21).**

| Background characteristics | Percentage (N) |
|---|---|
| **Assistance during delivery** | |
| Unskilled provider/No assistance | 9.8% (20,213) |
| Skilled provider | 90.2% (153,725) |
| **Antenatal care provider** | |
| Unskilled provider/No assistance | 14.9% (26,286) |
| Skilled provider | 85.1% (147,652) |
| **Ever experienced stillbirth/miscarriage** | |
| Yes | 74.6% (13,163) |
| No | 25.4% (4,357) |
| **Anemia** | |
| Others | 59.2% (98,116) |
| Not anemic | 40.8% (69,378) |
| **BMI** | |
| Others | 39.1% (62,841) |
| Normal | 60.9% (106,446) |
| **Birth weight of children** | |
| Low birth weight | 17.7% (26,796) |
| Normal | 82.3% (132,250) |
| **Delivery by caesarean section** | |
| Yes | 24.0% (37,265) |
| No | 76.0% (136,673) |

Source: Authors' calculations using NFHS (2019–2021).

some progress. However, there is notable progress in addressing anemia, with a considerable proportion of 40.8 per cent of women categorised as not anemic and more than half, 59.3 per cent, of severe, mild, and moderate anemia remaining concerning and warranting continued attention. Regarding BMI distributions, more than half of women fall within the normal range of 60.9 per cent. However, significant proportions of 39.1 are categorised as underweight, overweight, and obese. This diversity in BMI statuses highlights the complexity of nutritional challenges faced by women in India and calls for comprehensive approaches to address both undernutrition and overnutrition. Furthermore, the analysis reveals that 17.7 per cent of children are born with low birth weight, reflecting potential challenges in maternal and child health. Additionally, 24.0 per cent of deliveries occur by cesarean section, suggesting the need for further examination of maternal healthcare practices and interventions.

**Socio-economic background of RHI**

Table 3 presents the mean score of RHI by women's background characteristics. Women aged 15–24 years report the highest mean score of RHI 4.30, followed by those aged 25–34 (4.29), while women aged 35 and above show a slightly lower mea score (4.12), indicating better overall reproductive health among younger age compared to older age groups. Interestingly, the mean RHI score decreases as age increases, highlighting potential age-related health challenges, cumulative effects of multiple pregnancies, and reduced access to utilisation of maternal health services over time. Reproductive health outcomes improve with delayed marriage and childbearing. Women who married or had their first birth after age 18 report higher mean RHI scores (4.31 and 4.29, respectively) than those who did so earlier. The mean score is found to be higher among general groups (4.26) compared to Scheduled Castes (SC) and Scheduled Tribes (ST). There is a notable disparity in reproductive health among different religious groups, with Muslim women having a lower (4.25) compared to women of other religions. Economic status further influences women's RHI, with the wealth quintile demonstrating a significantly higher mean score. Women who have media exposure, even if partial, have a higher mean RHI score of 4.34, emphasising the importance of access to information. Furthermore, the Northern region shows the highest mean RHI score (4.42), followed by the Northeastern and Southern regions, whereas the Eastern region reports the lowest (4.09), highlighting persistent challenges. Moreover, women residing in urban areas displayed higher levels of RHI than those in rural areas, highlighting urban-rural inequalities in access to and quality of reproductive healthcare services.

Fig 1 indicates that married women with four or more children tend to have lower mean RHI levels than those with fewer children, particularly those with below replacement levels. This suggests that having fewer children is more favorable for women's overall health and well-being.

**District-level variation in women's reproductive health**

A district-wise mean score is analysed, and the result is depicted in Fig 2. Among the states, Rajasthan has the highest RHI score, followed by Kerala, Manipur, and Mizoram. In contrast, the lowest RHI score is observed in Jammu and Kashmir, followed by Punjab, Bihar, and Jharkhand. In nearly 15 states and union territories, the RHI score is below 4.35.

There is no state where the RHI score exceeds 5.25. In the Northern region, except for Jammu and Kashmir and Punjab, most states have moderate scores. On the other hand, in Central India, most of the states have lower levels of women's RHI, except Chhattisgarh. All states in the Southern region, except Telangana, demonstrate moderate to high RHI scores. Interestingly, the Western states of Maharashtra show a moderate to high score. However, the Eastern and northeastern states exhibit moderate to high RHI scores. Most North Eastern states like Mizoram, Manipur, and Tripura score high, while others secure moderate positions. The Eastern states, like West Bengal, Bihar, and Jharkhand, are notably weak, whereas states like Orissa are in the moderate range. These results indicate that most states should prioritise efforts to enhance women's reproductive health and improve their overall conditions.

**Table 3. Mean RHI by background characteristics among the women aged 15-49 in India (2019–21).**

| Background characteristics | Mean | SE | [95% Conf.Interval] | N |
|---|---|---|---|---|
| **Age** | | | | |
| 15-24 | 4.30 | 0.00 | [4.29-4.31] | 56621 |
| 25-34 | 4.29 | 0.00 | [4.28-4.29] | 102183 |
| 35+ | 4.12 | 0.01 | [4.10-4.13] | 16163 |
| **Age at marriage** | | | | |
| <18 | 4.21 | 0.00 | [4.20-4.22] | 56871 |
| >18 | 4.31 | 0.00 | [4.30-4.31] | 117930 |
| **Age at first birth** | | | | |
| <18 | 4.16 | 0.01 | [4.14-4.18] | 19765 |
| >18 | 4.29 | 0.00 | [4.28-4.30] | 155202 |
| **Social Groups** | | | | |
| SC | 4.23 | 0.01 | [4.22-4.25] | 39633 |
| ST | 4.23 | 0.01 | [4.22-4.25] | 17295 |
| OBC | 4.31 | 0.00 | [4.30-4.32] | 75241 |
| Others | 4.26 | 0.01 | [4.25-4.27] | 34256 |
| **Religion** | | | | |
| Hindu | 4.28 | 0.00 | [4.28-4.29] | 139221 |
| Muslim | 4.25 | 0.01 | [4.24-4.27] | 27845 |
| Christian | 4.33 | 0.02 | [4.30-4.37] | 3690 |
| Others | 4.13 | 0.02 | [4.10-4.16] | 4210 |
| **Mass media** | | | | |
| Not Exposed | 4.09 | 0.01 | [4.08-4.10] | 45992 |
| Partially exposed | 4.34 | 0.00 | [4.34-4.35] | 125777 |
| Exposed | 4.29 | 0.02 | [4.26-4.33] | 3198 |
| **Wealth quintiles** | | | | |
| Poorest | 4.06 | 0.01 | [4.05-4.07] | 39846 |
| Poorer | 4.29 | 0.01 | [4.28-4.30] | 36815 |
| Middle | 4.37 | 0.01 | [4.36-4.38] | 34253 |
| Richer | 4.37 | 0.01 | [4.36-4.39] | 33652 |
| Richest | 4.32 | 0.01 | [4.31-4.33] | 30399 |
| **Region** | | | | |
| North | 4.42 | 0.01 | [4.41-4.44] | 23776 |
| East | 4.09 | 0.01 | [4.08-4.10] | 45132 |
| Northeast | 4.38 | 0.01 | [4.36-4.41] | 7092 |
| Central | 4.29 | 0.01 | [4.28-4.30] | 46634 |
| West | 4.33 | 0.01 | [4.32-4.34] | 22609 |
| South | 4.35 | 0.01 | [4.33-4.36] | 29724 |
| **Residence** | | | | |
| Urban | 4.30 | 0.00 | [4.29-4.30] | 49346 |
| Rural | 4.27 | 0.00 | [4.26-4.27] | 125621 |

Source: Authors' calculations using NFHS (2019–2021).

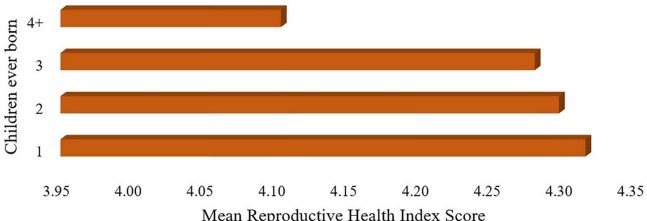

**Fig 1. Association between children ever born and Reproductive Health Index (RHI) score.**

### The association between women's reproductive health and background characteristics

The relationship between women's reproductive health and low fertility is complex and influenced by multiple background characteristics. The result of the Poisson regression model is presented in Table 4. It reveals a negative association between the number of CEB and reproductive health; women with higher CEB are less likely to have RHI, and the pattern is consistent even after adjusting the women's socio-economic and demographic characteristics. In Model 1, women with four or more children have significantly lower reproductive health scores than those with fewer children [IRR: 1.00; CI: 0.99, 1.00]. The trend persists in Model 2, even after adjusting for demographic and socio-cultural variables. Women with three or four children have significantly lower RHI scores compared to those with below replacement number of children [IRR: 0.97; CI: 0.96, 0.98]. Model 3 highlights that while the negative association between higher parity and reproductive health remains, mass media exposure and household wealth continue to show positive contributions to women's reproductive well-being, highlighting their potential role in supporting better health outcomes.

  Age is also a significant determinant, with older women (aged 35 and above) showing a negative association with RHI compared to younger age groups. Women aged 35 and above are significantly less likely to have higher reproductive health scores compared to those below the age of 24 [IRR: 0.97;CI: 0.96, 0.98]. Furthermore, women who married at 18 years or older have better reproductive health outcomes [IRR: 1.01;CI: 1.00, 1.01], while those who had their first birth at 18 years or older are less likely to have RHI scores [IRR: 1.01; CI: 1.00, 1.02]. Social groups exhibited differences in reproductive health, with ST [IRR: 1.02;CI: 1.01, 1.03] and Other Backward Class (OBC) [IRR: 1.01;CI: CI: 1.01, 1.02] being less likely to achieve high reproductive health compared to the general population. Religion affiliations also play a role, with Christians and individuals in the Others category having negative associations with RHI compared to Hindus. Mass media exposure and household wealth demonstrated a positive association with women's reproductive health. Women across wealth categories, from poorer [IRR: 1.05;CI: 1.04, 1.05] to richer [IRR: 1.05;CI: 1.04, 1.06], show better reproductive health outcomes. Partial exposure to mass media is significantly associated with higher RHI scores [IRR: 1.04;CI: 1.03, 1.05], indicating the importance of media in promoting reproductive health awareness. The Northeastern region shows a positive coefficient relative to the Northern region, though the association is not significant after adjustment. Lastly, RHI scores do not differ significantly between rural and urban areas in model 3.

### Discussion

The present study investigates the impact of declining fertility rates on women's reproductive health in India using the latest data from NFHS (2019–2021). The study reveals a negative association between having more children and RHI scores. Regression results also reflect a similar pattern, even after controlling for the demographic and social characteristics of the women. This result aligns with prior research indicating that higher parity often correlates with adverse health outcomes. For instance, studies have shown that having more children can increase the physical, emotional, and economic burdens on women [33,37,39,40]. Lower fertility rates can reduce these burdens, leading to better health outcomes

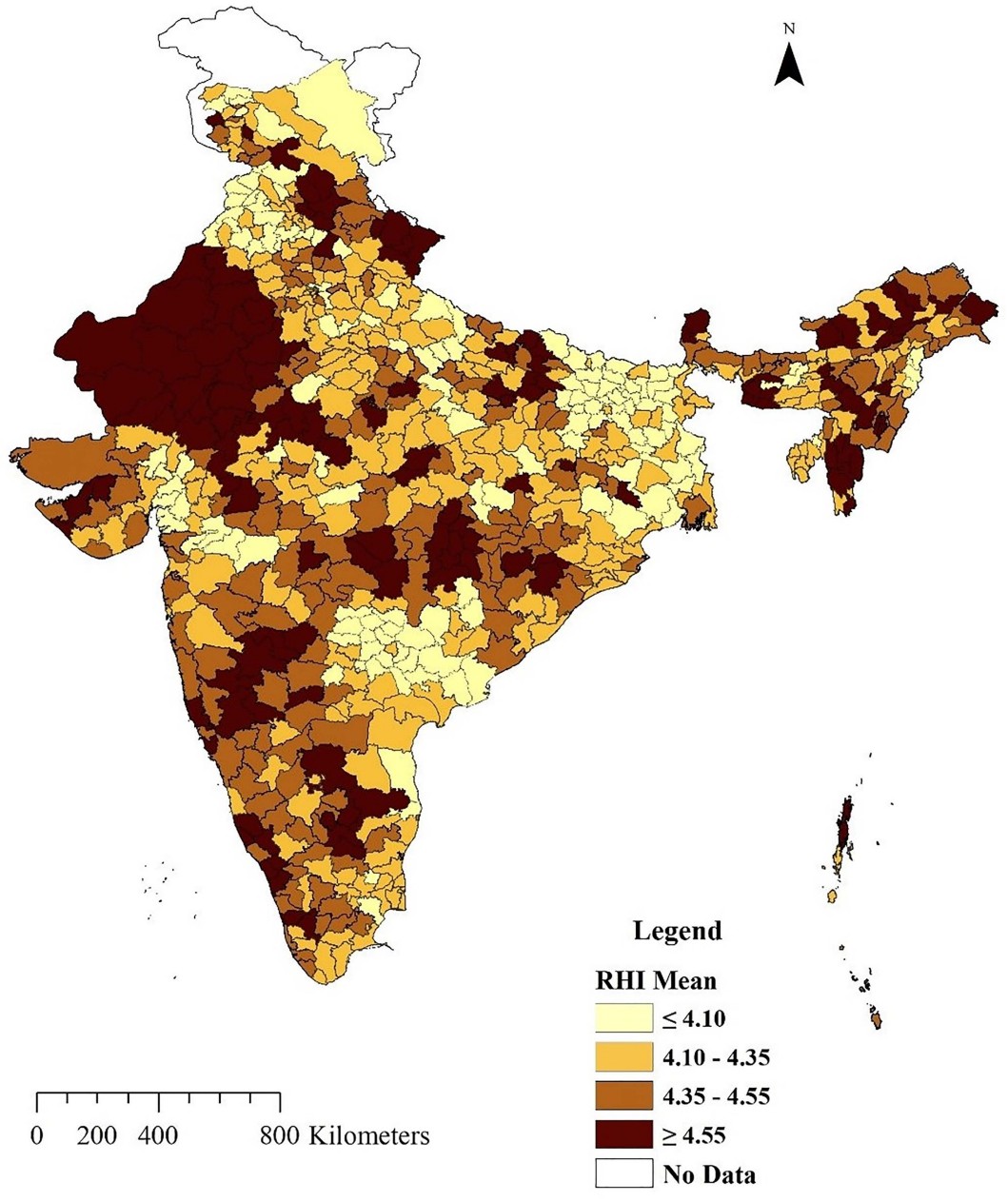

**Fig 2. District-level spatial distribution in Reproductive Health scoresacross India, 2019–21.**

and overall well-being [41]. However, this study uniquely highlighted that the benefits of low fertility are not uniformly distributed across all socio-economic and demographic groups in India.

The observed trends in antenatal care highlight the disparity in access to quality healthcare services, with a notable proportion of women receiving no antenatal care or care from unskilled providers. This is consistent with previous studies that have highlighted the inequalities in maternal health services in different groups [44]. In the same way, a substantial proportion of deliveries are attended by unskilled providers or without assistance. These findings underscore the

**Table 4. Poisson regression results (Incidence Rate Ratios) showing factors associated with women's reproductive health in India (2019–21).**

| RHI Score | Model 1 IRR with 95% CI | Model 2 IRR 95% CI | Model 3 IRR with 95% CI |
|---|---|---|---|
| **CEB** | | | |
| 1 | | | |
| 2 | 1.00 [0.99,1.00] | 1.00 [0.99,1.01] | 1.00 [1.00,1.01] |
| 3 | 0.99*** [0.98,0.99] | 1.00 [0.99,1.01] | 1.00 [1.00,1.01] |
| 4+ | 0.95*** [0.94,0.96] | 0.97*** [0.96,0.98] | 0.98*** [0.97,0.99] |
| **Age** | | | |
| 15-24 | | | |
| 25-34 | | 1.00 [0.99,1.00] | 0.99 [0.99,1.00] |
| 35+ | | 0.97*** [0.96,0.98] | 0.97*** [0.96,0.98] |
| **Age at marriage** | | | |
| <18 | | | |
| >18 | | 1.01** [1.00,1.02] | 1.01 [1.00,1.01] |
| **Age at first birth** | | | |
| <18 | | | |
| >18 | | 1.01** [1.00,1.02] | 1.01** [1.00,1.02] |
| **Social Groups** | | | |
| SC | | | |
| ST | | 1.01 [1.00,1.02] | 1.02*** [1.01,1.03] |
| OBC | | 1.02*** [1.01,1.03] | 1.01*** [1.01,1.02] |
| Others | | 1.00 [1.00,1.01] | 1.00 [0.99,1.01] |
| **Religion** | | | |
| Hindu | | | |
| Muslim | | 0.99* [0.98,1.00] | 0.99 [0.99,1.00] |
| Christian | | 1.00 [0.99,1.01] | 0.99 [0.98,1.01] |
| Others | | 0.96*** [0.95,0.98] | 0.96*** [0.95,0.97] |
| **Mass media** | | | |
| Not Exposed | | | |
| Partially exposed | | | 1.04*** [1.03,1.05] |
| Exposed | | | 1.03** [1.01,1.05] |
| **Wealth quintiles** | | | |
| Poorest | | | |
| Poorer | | | 1.05*** [1.04,1.05] |
| Middle | | | 1.06*** [1.05,1.06] |
| Richer | | | 1.05*** [1.04,1.06] |
| Richest | | | 1.03*** [1.02,1.04] |
| **Region** | | | |
| North | | | |
| East | | 0.94*** [0.93,0.95] | 0.96*** [0.95,0.96] |
| Northeast | | 1.01 [1.00,1.02] | 1.01* [1.00,1.02] |
| Central | | 0.98*** [0.97,0.98] | 0.99*** [0.98,0.99] |
| West | | 0.98*** [0.97,0.99] | 0.98** [0.97,0.99] |
| South | | 0.97*** [0.97,0.98] | 0.97*** [0.96,0.97] |
| **Residence** | | | |
| Urban | | | |
| Rural | | 1.00 [0.99,1.00] | 1.00 [1.00,1.01] |

Source: Authors' calculations using NFHS (2019–2021).

importance of enhancing healthcare infrastructure and accessibility to ensure comprehensive maternal and reproductive health services for women nationwide [45].

The persistent challenges in maternal health and nutrition are evident from the high prevalence of stillbirths, miscarriages, and anemia among women. This aligns with previous studies indicating that adverse pregnancy outcomes are often linked to inadequate prenatal care and socioeconomic disparities [46,47]. Despite progress in addressing anemia, the study highlights the continued prevalence of moderate and mild cases. Previous research shows that anemia is often linked to poor dietary intake, high fertility rates, and low socioeconomic status of women [48]. Addressing anemia requires comprehensive nutritional interventions and improved healthcare services.

The diversity of BMI distributions reflects the complex nutritional challenges women face in India, necessitating comprehensive approaches to address both undernutrition and overnutrition. Previous studies have highlighted the dual burden of malnutrition in India [42,49,50]. The prevalence of low birth weight among children highlights significant concerns in maternal and child health, which are often associated with inadequate maternal nutrition and healthcare during pregnancy [51]. The relatively high rate of caesarean deliveries suggests the need for further examination of maternal healthcare practices and potential over-medicalisation of childbirth, which has been observed in other studies as well [52].

Furthermore, the analysis reveals negative associations between women's reproductive health and various demographic and socio-economic factors such as older age, early marriage, and higher parity, highlighting the importance of reproductive choices and healthcare access. Disparities based on social groups, religion, media exposure, wealth quintiles, and regional variations underscore the need for targeted interventions to address inequalities and promote women's reproductive health status. Notably, some districts in the Northeastern region report higher RHI scores despite socio-economic disadvantages, reflecting stronger community engagement in maternal care, particularly in tribal and ethnic regions where cultural practices and outreach-based health services contribute positively to reproductive health [41]. The regression analysis reveals that rural residency negatively affects women's reproductive health, but this factor can be mitigated by economic well-being and media exposure. In rural India, nearly half of the women did not have regular exposure to any form of media, whereas in urban areas, it was less [11]. Rural women have less exposure to mass media, emphasising the need for targeted interventions to improve healthcare access, enhance nutritional status, and address socio-economic disparities.

Therefore, the findings emphasise the importance of holistic approaches to address the complex interplay of factors influencing women's reproductive health and overall well-being. Targeted interventions to improve healthcare access, enhance nutritional status, and address socio-cultural disparities are essential for promoting gender equality and fostering sustainable development in India. By integrating these insights into policy formulation and program design, policymakers and stakeholders can work towards ensuring their rights to health and well-being.

## Limitations and strengths of the study

This study has some limitations that need to be acknowledged. Firstly, the cross-sectional design of the DHS and NFHS data limits the ability to find a causal relationship between fertility levels and reproductive health outcomes. While cross-sectional data allows for identifying association, the temporal effects cannot be firmly established. Secondly, some of the variables are self-reported data, which may introduce recall and reporting bias. Thirdly, despite the controlling for numerous socio-economic and demographic factors, the unmeasured confounding factors such as cultural norms and informal healthcare practices may influence the result.

Despite these limitations, the study has several strengths. It uses nationally representative data from NFHS-5 to ensure that the findings are robust and enhance the reliability of the results. Moreover, the study adopts the capability approach, health inequity and life cycle approach to understand the disparities across socio-economic, demographic, and regional groups. Finally, the study's policies focus on the vulnerable groups and equity issues and offer critical insights for targeted interventions. These findings inform policies that promote the improvement of reproductive health, gender equality, and sustainable societal development in India's diverse socio-cultural landscape.

## Conclusions and policy implications

The findings of this study provide a critical foundation for understanding the relationship between fertility transitions and women's reproductive health in India, emphasising the significant variations across different socio-economic and demographic groups that influence health outcomes. While fertility transition in India is associated with improved maternal health, the benefits remain unequally distributed, particularly among disadvantaged backgrounds, including those from SC, ST and economically weaker populations. Women from higher socio-economic backgrounds and urban areas tend to experience more substantial improvements in their reproductive health, whereas those from lower socio-economic groups and rural areas face persistent challenges. Additionally, the study highlights the importance of considering regional variations in health policies to ensure that the benefits of lower fertility are accessible to women across diverse settings in India.

Addressing these disparities requires a comprehensive policy approach that strengthens healthcare infrastructure, ensures equitable access to maternal healthcare services, and enhances reproductive health awareness. A key policy should be taken to improve healthcare access in underperforming regions by expanding the availability of skilled birth attendants, integrating mobile health units in backward rural areas, and promoting telemedicine-based consultations for antenatal and postnatal care. Additionally, conditional cash transfer like Janani Suraksha Yojana (JSY) can increase the institutional delivery and be linked to antenatal care visit, can incentivise utilisation of maternal health services among low income groups. Strengthening nutritional support programs is essential to reduce anemia, malnutrition and low birth weight through food fortification policies, iron and folic acid supplementation, and maternal nutrition counseling. Additionally, the overuse of cesarean deliveries in private healthcare must be regulated through governmental policies, and the number of midwife-led birthing centers must be increased to encourage safe childbirth practices. Expanding the role of community health workers to engage with these populations can further enhance service utilisation and awareness regarding reproductive health services. Simultaneously, state-led mass media campaigns are essential to improve birth spacing, contraceptive use, and safe delivery practices and also increase reproductive health awareness.

This research contributes to the broader discourse on fertility management and reproductive health by emphasising the importance of ensuring women's health and well-being. Aligning these efforts with SDG goals, especially SDG 3 (Good Health and Well-being) and SDG 5 (Gender Equality), will be essential to promote gender-responsive health policies. Addressing all the constraints and enhancing equitable healthcare access for all will be essential for translating the benefits of low fertility into improved reproductive health and broader societal development.

## Supporting information

**S1 Table. Sensitivity analysis using Poisson regression models based on alternate Reproductive Health Index (RHI) specifications.**
(DOCX)

## Acknowledgments

This paper was presented at the 35th International Geographical Congress (IGC), 2024, in Dublin, Ireland. We are grateful for the valuable feedback received from experts during the conference, which has been incorporated into this study and has significantly improved the quality of the manuscript. We thank the anonymous reviewers for their valuable comments and constructive feedback, which significantly improved the clarity and quality of the paper.

## Author contributions

**Conceptualization:** Roni Sikdar, Dhananjay W. Bansod.

**Data curation:** Roni Sikdar.

**Formal analysis:** Roni Sikdar.

**Methodology:** Roni Sikdar, Dhananjay W. Bansod.

**Supervision:** Dhananjay W. Bansod.

**Validation:** Dhananjay W. Bansod.

**Writing – original draft:** Roni Sikdar.

**Writing – review & editing:** Roni Sikdar, Dhananjay W. Bansod.

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
