## [Decision Letter · Decision Letter 0]

10 Mar 2025

PONE-D-25-01359Does Low Fertility Indicate Better Reproductive Health Status? Evidence from Nationally Representative Survey in IndiaPLOS ONE?

Dear Dr. Sikdar,

Thank you for submitting your manuscript to PLOS ONE. After careful consideration, we feel that it has merit but does not fully meet PLOS ONE’s publication criteria as it currently stands. Therefore, we invite you to submit a revised version of the manuscript that addresses the points raised during the review process.

We look forward to receiving your revised manuscript.

Kind regards,

Pijush Kanti Khan, Ph.D.

Academic Editor

PLOS ONE

Journal Requirements:

3. We note that Figure 2 in your submission contain [map/satellite] images which may be copyrighted. All PLOS content is published under the Creative Commons Attribution License (CC BY 4.0), which means that the manuscript, images, and Supporting Information files will be freely available online, and any third party is permitted to access, download, copy, distribute, and use these materials in any way, even commercially, with proper attribution. For these reasons, we cannot publish previously copyrighted maps or satellite images created using proprietary data, such as Google software (Google Maps, Street View, and Earth). For more information, see our copyright guidelines: http://journals.plos.org/plosone/s/licenses-and-copyright.

1. You may seek permission from the original copyright holder of Figure 2 to publish the content specifically under the CC BY 4.0 license. 

Reviewers' comments:

Reviewer's Responses to Questions

**Comments to the Author**

1. Is the manuscript technically sound, and do the data support the conclusions?

Reviewer #1: Yes

Reviewer #2: Yes

Reviewer #3: Yes

Reviewer #4: No

2. Has the statistical analysis been performed appropriately and rigorously?

Reviewer #1: Yes

Reviewer #2: Yes

Reviewer #3: No

Reviewer #4: Yes

3. Have the authors made all data underlying the findings in their manuscript fully available?

Reviewer #1: Yes

Reviewer #2: Yes

Reviewer #3: Yes

Reviewer #4: Yes

4. Is the manuscript presented in an intelligible fashion and written in standard English?

Reviewer #1: Yes

Reviewer #2: Yes

Reviewer #3: No

Reviewer #4: Yes

Reviewer #1: The paper has novelty and significance in respect to current women health condition in India. But this paper has been analysed and written very well but it has some structural problem that need to improve for better readability of the paper. Here are my following suggestions.

1.Line 45: The authors have written (coefficient = -0.05; CI: -0.06, -0.05), what is the relevance of these values in abstract, I would suggest to remove the values from abstract and only focus on the outcome of analysis.

2.I do not feel the necessary of separate Introduction and Fertility Transition and Reproductive Health section, rather I would suggest to integrate them and strengthen the background of the study.

3.Do you really need this sentence in line 162-163- “The International Institute for Population Sciences (IIPS) in Mumbai conducted the survey with funding from the Government of India's Ministry of Health and Family Welfare”. I don’t feel so. Similar for line 166-167. Streamline the line 161-168, and remove unnecessary lines.

4.After reading the whole methodology I feel that it needs to rearrange, as my previous comment, remove unnecessary writing for data, sample size and Variables and their measurement section. Combine them together in one section, and named the section- ‘study variables, measurements and sources’ something like that. Then in that section first paragraph write data source and samples, then onwards variable measurements.

5.Line 215: Do you want to cite STATA, if not then what is the need for writing this-(Stata Corporation, College Station, TX), as you already stated that you used STATA 17.

6.I would Like to give citation or equation in like 202-206

7.-I would like to give regression equation forms of two poison regression.

8.Line 2015-2017- you wrote that “We use ArcMap for creating maps. The shape file for the maps is downloaded from the official Demographic Health Survey (https://dhsprogram.com/data/available-datasets.cfm) website”. If you really feel that you need to mentioned this, concise the sentences.

9.In table 1: what is the meaning of given two collum for category ? instead you simple in one column, write category and in bracket yes/no (category (yes/no)), is not it represent the same as you did. Think. What you did makes the paper structure unesthetic and reader will lose the concentration while reading.

10.Table 2: similar problem like table one, you can simple write totalvalue and % in bracket- 20,213 (9.8).

11.I do not understand the meaning of “No one” in table 2, can you clarify and reframe.

12.In table 2. For Ever experienced Still Birth/miscarriage- you have written separate percentage for yes and for no, together 100%. Then what is the meaning of given both yes and no, if 74.56 % is for yes, then rest of the % will by default become no. can you explain?

13.In line 246- you write a heading of a section “Mean number of RHI in background characteristics-wise”, I won’t do that, mean is the analysis you did, it doesnot represent the content of the section. Rather write “socio-economic/ecological/political/cultural background of RHI”

14.Which one is figure 1?, author should have basic responsibility to frame the paper.

15.I don’t feel the necessity of writing composite score in the title “District-level Variation in Women’s Reproductive Health Composite Score”

16.Where is figure 2 ?

17.Table 4, what the value inside table represents? standard error? or coefficient? or something else. How do I know? This reflects the lack of professionalism by the authors. I would suggest to follow good article, and carefully check how they represent the regression table in the paper.

18.Your policy suggestions are too generals, based on your study, What policy measure will you take if you become the sole responsible authority.

19.Overall, I feel paper is lengthy, and some part of the writing is exaggerated and over romanticised.

Over all paper is interesting, I feel this structural change make the paper more concise and increase the understandability and readability of the non-subject readers also. I recommend the paper for above minor, not major revision.

Reviewer #2: The manuscript presents a rigorously conducted study characterised by a high level of technical competence. The data is robust, and the conclusions drawn are substantiated by the experiments and analyses executed. The statistical analysis is both appropriate and comprehensive, which enhances the reliability of the findings and their interpretation. Furthermore, the authors have made all underlying data available in accordance with PLOS Data policies, thereby ensuring transparency. The manuscript is well-structured and articulately written in standard English, rendering the content both accessible and easily comprehensible.

Reviewer #3: This cross-sectional study attempts to explore the association between declining fertility rates and women's reproductive health in India, considering socioeconomic and demographic factors as well as regional variations. The objective of this paper is quite relevant & interesting, but has some shortcomings which needs major revisions & rework. Hence, upon preliminary review, my recommendation is for a major revision before considering the manuscript for publication.

Followings are some of my observations that needs further clarification and revision:

1.The article requires substantial revision to address grammatical problems and repetitive sentences (e.g., lines 355-359). Furthermore, the article appears to be very lengthy, and the assertions in the findings, discussions, and conclusion sections are very repeated.

2.Line no. 106-107: ‘By integrating these …….low fertility on holistic well-being”. I have a serious concern in using the term ‘holistic well-being of women’ which can include both mental and social well-being as well (as the author mentioned in 99-101). But in the analysis, there are no indicators on social or mental well being of women. As a result, it is better to avoid using this term or to explain how your seven indicators can justify mental and social well-being.

3.Line no. 224-25: ‘However, 9.78% of deliveries ……remains insufficient.’ Lower percentages of deliveries by skilled persons does not always imply lower access to health care facilities. What about the families who don’t opt for such deliveries and prefer delivery at home?

4.Line no. 258-259: ‘Regarding regional variations…..RHI’. Even northern regions have highest mean RHI (4.42). But the author doesn’t mention that. Additionally, interpretation of regression mentioned (in line no. 316-318) north-eastern region showing lower likelihood of RHI, while the values are positive. Please rework and rewrite the section.

5.In some cases, the OR values are very low. Plus, the author needs to mention the high or low RHI score range as the scores does not vary much (such as 4.33 in West and 4.35 in South). Please elaborate on how far it is justified to say that regional disparities exist (line 316-318)?

6.Line 318-319: ‘Additionally, residence plays…. counterparts (Model 3)’. In model 3 Rural and urban odds are not significant. Therefore, no difference between rural and urban areas.

7.Line 297-300: ‘Model 3 highlights….wealth index’. Please elaborate, as the models do not show that.

8.Please recheck the Poisson regression model. Odds ratio can’t be negative. An odds ratio greater than 1 indicates an increased risk associated with the exposure, while an odds ratio less than 1 indicates a decreased risk. Do the values in Table 4 represent odds or log odds? If log odds, then the interpretation should alter.

9.The author tries to investigate the spatial variation. What about the role of clustering effect on depended variable due to hierarchical nature of NFHS data? Can Poisson regression model consider that effect?

Reviewer #4: 1.The title of the paper does not reflect the analysis presented; the author does not analyse low fertility but rather focuses on the factors associated with women’s reproductive health.

2.The abstract of the paper does not follow the journal's format.

3.The introduction section does not clearly mention the research gap and the objective of the paper.

4.The concept of the Reproductive Health Index (RHI) is not clearly defined in the introduction. The benefits and how NFHS data can be used to measure this index should be explained more clearly.

5.The limitations of the paper should be included in the discussion section, not the introduction section.

6.The justification for using the Poisson regression model should be clarified, especially over other models such as the negative binomial regression.

7.The justification for the equal weighting of indicators in the RHI should be explained. The author could conduct a sensitivity analysis to support this approach.

8.The Northeastern region is associated with higher RHI scores despite socioeconomic disadvantages; this finding needs more discussion.

**Do you want your identity to be public for this peer review?** For information about this choice, including consent withdrawal, please see our Privacy Policy

Reviewer #1: No

Reviewer #2: **Yes: ** Papai Barman

Reviewer #3: No

Reviewer #4: No

---

## [Author Response · Author response to Decision Letter 1]

12 Apr 2025

Reviewer #1: The paper has novelty and significance in respect to current women health condition in India. But this paper has been analysed and written very well but it has some structural problem that need to improve for better readability of the paper. Here are my following suggestions.

Reply: We are sincerely grateful for the time and efforts for the reviews of our manuscript, “Does Low Fertility Indicate Better Reproductive Health Status? Evidence from Nationally Representative Survey in India.” We appreciate the constructive feedback, which has helped us to strengthen the manuscript significantly.

Comment 1. Line 45: The authors have written (coefficient = -0.05; CI: -0.06, -0.05), what is the relevance of these values in abstract, I would suggest to remove the values from abstract and only focus on the outcome of analysis.

Reply: Thank you. We have removed the coefficient values and retained only the outcomes in the abstract.

Comment 2. I do not feel the necessary of separate Introduction and Fertility Transition and Reproductive Health section, rather I would suggest to integrate them and strengthen the background of the study.

Reply: We appreciate this suggestion. These sections have been merged and the background has been strengthened (see lines 97–120 and 137–139).

Comment 3. Do you really need this sentence in line 162-163- “The International Institute for Population Sciences (IIPS) in Mumbai conducted the survey with funding from the Government of India's Ministry of Health and Family Welfare”. I don’t feel so. Similar for line 166-167. Streamline the line 161-168, and remove unnecessary lines.

Reply: Thank you for this suggestion. We addressed this comment and removed all the lines from the data as suggested.

Comment 4. After reading the whole methodology I feel that it needs to rearrange, as my previous comment, remove unnecessary writing for data, sample size and Variables and their measurement section. Combine them together in one section, and named the section- ‘study variables, measurements and sources’ something like that. Then in that section first paragraph write data source and samples, then onwards variable measurements.

Reply: Addressed. We have created a two section titled Data Sources and Study Variables and their Measurements, combining the data, variables, and source sections.

Comment 5. Line 215: Do you want to cite STATA, if not then what is the need for writing this-(Stata Corporation, College Station, TX), as you already stated that you used STATA 17.

Reply: Thank you for this suggestion. The citation has been removed.

Comment 6. I would Like to give citation or equation in like 202-206

Reply: We have added the composite index equation and appropriate citations (see lines 180–183).

Comment 7. I would like to give regression equation forms of two poison regressions.

Reply: Thank you for this suggestion. We have addressed the comment and added a regression equation in the methodology section (lines 186-191).

Comment 8. Line 2015-2017- you wrote that “We use ArcMap for creating maps. The shape file for the maps is downloaded from the official Demographic Health Survey (https://dhsprogram.com/data/available-datasets.cfm) website”. If you really feel that you need to mentioned this, concise the sentences.

Reply: Addressed. The sentence has been shortened, and the data source part has been removed.

Comment 9. In table 1: what is the meaning of given two column for category ? instead you simple in one column, write category and in bracket yes/no (category (yes/no)), is not it represent the same as you did. Think. What you did makes the paper structure unesthetic and reader will lose the concentration while reading.

Reply: Thank you for your valuable feedback. We acknowledge that having two separate columns for categories may affect the readability of the table. Based on your suggestion, we have reformatted Table 1 with a single column for each variable with values (Yes/No) to improve readability and aesthetics.

Comment 10. Table 2: similar problem like table one, you can simple write total value and % in bracket- 20,213 (9.8).

Reply: Thank you for your comment. We have deleted the second percentage column and merged the total values and percentages into a single column (e.g., 20,213 (9.8%)) to improve readability and ensure consistency across all categories.

Comment 11. I do not understand the meaning of “No one” in table 2, can you clarify and reframe.

Reply: We have replaced “No one” with “Unskilled provider/No assistance” to clarify that this category includes cases where women either received assistance from an unskilled provider or had no assistance.

Comment 12. In table 2. For Ever experienced Still Birth/miscarriage- you have written separate percentage for yes and for no, together 100%. Then what is the meaning of given both yes and no, if 74.56 % is for yes, then rest of the % will by default become no. can you explain?

Reply: We appreciate this comment. We have removed the “No” row and retained only the 'Yes' category. A footnote has been added for clarity.

Comment 13. In line 246- you write a heading of a section “Mean number of RHI in background characteristics-wise”, I won’t do that, mean is the analysis you did, it does not represent the content of the section. Rather write “socio-economic/ecological/political/cultural background of RHI”

Reply: We have removed the line Mean number of RHI and addressed this comment by adding the ‘Socio-economic background of RHI’.

Comment 14. Which one is figure 1?, author should have basic responsibility to frame the paper.

Reply: Thank you for your suggestion. Figures are provided separately in TIFF format as per journal submission guidelines.

Comment 15. I don’t feel the necessity of writing composite score in the title “District-level Variation in Women’s Reproductive Health Composite Score”

Reply: The title has been revised as suggested.

Comment 16. Where is figure 2?

Reply: The figure is provided separately in the TIFF file as per the journal’s guidelines.

Comment 17. Table 4, what the value inside table represents? standard error? or coefficient? or something else. How do I know? This reflects the lack of professionalism by the authors. I would suggest to follow good article, and carefully check how they represent the regression table in the paper.

Reply: Thank you for this suggestion. The table now included coefficients and confidence intervals.

Comment 18. Your policy suggestions are too generals, based on your study, What policy measure will you take if you become the sole responsible authority.

Reply: The policy implication section has been revised to reflect precise and actionable recommendations derived from the findings of the study.

Comment 19. Overall, I feel paper is lengthy, and some part of the writing is exaggerated and over romanticised.

Reply: Thank you. We have revised the introduction and reduced redundancy to improve conciseness.

Over all paper is interesting, I feel this structural change make the paper more concise and increase the understandability and readability of the non-subject readers also. I recommend the paper for above minor, not major revision.

Reply: Thank you. The manuscript has been revised accordingly for enhanced readability and organisation.

Reviewer #2: The manuscript presents a rigorously conducted study characterised by a high level of technical competence. The data is robust, and the conclusions drawn are substantiated by the experiments and analyses executed. The statistical analysis is both appropriate and comprehensive, which enhances the reliability of the findings and their interpretation. Furthermore, the authors have made all underlying data available in accordance with PLOS Data policies, thereby ensuring transparency. The manuscript is well-structured and articulately written in standard English, rendering the content both accessible and easily comprehensible.

Reply: Thank you for your supportive feedback. We appreciate your positive evaluation.

Reviewer #3: This cross-sectional study attempts to explore the association between declining fertility rates and women's reproductive health in India, considering socioeconomic and demographic factors as well as regional variations. The objective of this paper is quite relevant & interesting, but has some shortcomings which needs major revisions & rework. Hence, upon preliminary review, my recommendation is for a major revision before considering the manuscript for publication.

Reply: We sincerely thank the reviewer for acknowledging the relevance and importance of our study. We have carefully addressed all suggested revisions to improve the clarity, structure, and interpretation. We hope the revised manuscript meets the expectations for further consideration.

Followings are some of my observations that needs further clarification and revision:

Comment 1. The article requires substantial revision to address grammatical problems and repetitive sentences (e.g., lines 355-359). Furthermore, the article appears to be very lengthy, and the assertions in the findings, discussions, and conclusion sections are very repeated.

Reply: Thank you for your observation. We have thoroughly revised the manuscript to correct grammatical errors and improve clarity. We have deleted the repetitive sentences from the findings and discussion.

Comment 2. Line no. 106-107: ‘By integrating these …….low fertility on holistic well-being”. I have a serious concern in using the term ‘holistic well-being of women’ which can include both mental and social well-being as well (as the author mentioned in 99-101). But in the analysis, there are no indicators on social or mental well being of women. As a result, it is better to avoid using this term or to explain how your seven indicators can justify mental and social well-being.

Reply: We agree with the concern. The phrase “holistic well-being” has been replaced with more precise terms, such as “reproductive health,” in relevant sections to maintain objectivity.

Comment 3. Line no. 224-25: ‘However, 9.78% of deliveries ……remains insufficient.’ Lower percentages of deliveries by skilled persons does not always imply lower access to health care facilities. What about the families who don’t opt for such deliveries and prefer delivery at home?

Reply: Thank you for this important observation. We agree that lower rates of skilled delivery attendance may not always reflect limited access to healthcare services. Cultural preferences, traditional beliefs, and personal choices can also influence the decision to deliver at home without skilled assistance. To address this, we have revised the statement to acknowledge that while the proportion of unskilled deliveries may indicate service gaps, it may also reflect the voluntary preference for home births in certain communities.

Comment 4. Line no. 258-259: ‘Regarding regional variations…..RHI’. Even northern regions have highest mean RHI (4.42). But the author doesn’t mention that. Additionally, interpretation of regression mentioned (in line no. 316-318) north-eastern region showing lower likelihood of RHI, while the values are positive. Please rework and rewrite the section.

Reply: Thank you for the comment. You are correct that the Northern region shows the highest mean RHI score in the descriptive analysis. However, the difference between the North and North Eastern regions becomes statistically insignificant in the regression after adjusting for covariates. Therefore, we rely on the descriptive means for interpretation in this case. Also, we rewrite the section mentioned above.

Comment 5. In some cases, the OR values are very low. Plus, the author needs to mention the high or low RHI score range as the scores does not vary much (such as 4.33 in West and 4.35 in South). Please elaborate on how far it is justified to say that regional disparities exist (line 316-318)?

Reply: Thank you for pointing this out. We have corrected the earlier error where coefficients were mistakenly referred to as odds ratios. These are now properly labeled as Poisson regression coefficients. As for the regional variation, we have included the actual RHI score range (4.09 to 4.42) in the results for clarity. Since the difference is slight and the North Eastern region’s coefficient is not statistically significant, to avoid overstatement, we removed the line referencing regional disparities.

Comment 6. Line 318-319: ‘Additionally, residence plays…. counterparts (Model 3)’. In model 3 Rural and urban odds are not significant. Therefore, no difference between rural and urban areas.

Reply: Thank you for pointing this out. We agree that the difference between rural and urban residence is not statistically significant in Model 3. We have revised the sentence in the results section.

Comment 7. Line 297-300: ‘Model 3 highlights….wealth index’. Please elaborate, as the models do not show that.

Reply: Thank you for this helpful observation. We agree that the original interpretation overstated the role of mass media and wealth index in mitigating the effects of higher parity. We have revised the sentence to reflect the results of Model 3 more accurately.

Comment 8. Please recheck the Poisson regression model. Odds ratio can’t be negative. An odds ratio greater than 1 indicates an increased risk associated with the exposure, while an odds ratio less than 1 indicates a decreased risk. Do the values in Table 4 represent odds or log odds? If log odds, then the interpretation should alter.

Reply: Thank you for pointing this out. You are absolutely correct; odds ratios cannot be negative. In the original manuscript, we mistakenly referred to Poisson regression coefficients as odds ratios (OR), which was inaccurate. The values presented in Table 4 are actually log-linear coefficients, not odds ratios. We have corrected the terminology throughout the results and discussion sections. All interpretations have been revised accordingly to reflect the direction and magnitude of association in terms of coefficients, not ORs.

Comment 9. The author tries to investigate the spatial variation. What about the role of clustering effect on depended variable due to hierarchical nature of NFHS data? Can Poisson regression model consider that effect?

Reply: Thank you for your comment. It is also true that NFHS has used multistage stratified clustering sampling; it already adjusted the hierarchical nature of the dataset using the design effect. Therefore, I believe that the use of Poisson regression in this dataset is suitable.

Reviewer #4: 1. The title of the paper does not reflect the analysis presented; the author does not analyse low fertility but rather focuses on the factors associated with women’s reproductive health.

Reply: Thank you for your feedback. We agree that the analysis primarily focuses on women's reproductive health, but this is intentionally framed within the broader context of India's ongoing fertility transition. The study aims to explore how reproductive health outcomes are shaped during a period of declining fertility, which justifies the use of “low fertility” in the title. We believe the current title captures the thematic intent of the paper and its relevance to the demographic shift under study.

Comment 2. The abstract of the paper does not follow the journal's format.

Reply: Thank you for your observation. We have revised the abstract into different sections, such as the introduction, methods, results, and conclusion.

Comment 3. The introduction section does not clearly mention the research gap and the objective of the paper.

Reply: Thank you for your comment. We have revised the introduction to clearly state the research gap and objectives of the paper. This has been addressed in lines 125-132 of the revised manuscript.

Comment 4. The concept of the Reproductive Health Index (RHI) is not clearly defined in the introduction. The benefits and how NFHS data can be used to measure this index should be explained more clearly.

Reply: Thank you for this helpful comment. We have now clarified the concept of the Reproductive Health Index (RHI) in the introduction (128-130). A more detailed explanation of the selected NFHS indicators and how the index is constructed is provided in the methodology sec

---

## [Decision Letter · Decision Letter 1]

1 Jul 2025

PONE-D-25-01359R1Does Low Fertility Indicate Better Reproductive Health Status? Evidence from Nationally Representative Survey in IndiaPLOS ONE?

Dear Dr. Sikdar,

Thank you for submitting your manuscript to PLOS ONE. After careful consideration, we feel that it has merit but does not fully meet PLOS ONE’s publication criteria as it currently stands. Therefore, we invite you to submit a revised version of the manuscript that addresses the points raised during the review process.

We look forward to receiving your revised manuscript.

Kind regards,

Pijush Kanti Khan, Ph.D.

Academic Editor

PLOS ONE

Journal Requirements:

Additional Editor Comments:

Dear Authors,

Thank you for your revised manuscript. The paper has substantially improved after the revision, and I appreciate the effort you’ve put into addressing the earlier comments. That said, I believe there are still a few minor issues that need to be addressed before the paper is ready for publication. Therefore, I would like to request a minor revision. Please revise the manuscript accordingly and resubmit it by 15th July, 2025. Feel free to reach out if you have any questions or need further clarification. Thank you,

Best regards

Reviewers' comments:

Reviewer's Responses to Questions

**Comments to the Author**

Reviewer #3: All comments have been addressed

Reviewer #4: (No Response)

2. Is the manuscript technically sound, and do the data support the conclusions?

Reviewer #3: Yes

Reviewer #4: No

3. Has the statistical analysis been performed appropriately and rigorously?

Reviewer #3: Yes

Reviewer #4: I Don't Know

4. Have the authors made all data underlying the findings in their manuscript fully available?

Reviewer #3: Yes

Reviewer #4: Yes

5. Is the manuscript presented in an intelligible fashion and written in standard English?

Reviewer #3: Yes

Reviewer #4: No

Reviewer #3: The article addresses a crucial aspect of women's reproductive health and its association with low fertility rates in India. This topic remains relatively underexplored, making the study particularly relevant in the Indian context. After an initial round of revisions, the manuscript is now well-organized and clearly written. The following are some minor suggestions that may further improve clarity:

1. Kindly assess the level of multicollinearity—particularly among variables such as age, age at marriage, and age at first birth—prior to conducting the regression analysis, and include the findings in the manuscript.

2. Please examine the potential presence of endogeneity, as certain variables used in constructing the Reproductive Health Index (e.g., history of stillbirth/miscarriage, type of delivery assistance) may influence the number of children ever born (CEB). A rationale for the absence of endogeneity in the analysis should be provided.

3. Specify the states included in each of the six regional categories, or cite the source from which the regional classification was derived.

4. In the methodology section, please justify the use of the Poisson regression model. Also, check for the presence of overdispersion or under dispersion in the data and report the findings.

5. It is recommended to present Relative Risk Ratios (RRR) in Poisson regression results instead of regression coefficients for clearer interpretation of the results.

6. Ensure that all figures have appropriate titles and are properly numbered.

Reviewer #4: 1.The introduction section includes limited literature; the authors need to add more recent studies.

2.The authors should provide more information about the NFHS data, including the questions asked, which questions were used in this study, and the topics covered by NFHS.

3. The study variables and their measurement methods require further details.

4.In Table 2, the authors report that 75% of respondents have ever experienced a stillbirth or miscarriage. This figure appears inaccurate, as the NFHS report indicates only 7.3% for miscarriage and 0.9% for stillbirth. The authors should verify these figures.

5.In Table 3, the mean RHI varies by women’s age, age at first marriage, and age at first birth. While RHI increases with higher age at first marriage and first birth, it appears to decrease with advancing age of women. This pattern needs further explanation from the authors to clarify the underlying reasons.

**Do you want your identity to be public for this peer review?** For information about this choice, including consent withdrawal, please see our Privacy Policy

Reviewer #3: No

Reviewer #4: No

---

## [Author Response · Author response to Decision Letter 2]

4 Jul 2025

Reviewer #3: The article addresses a crucial aspect of women's reproductive health and its association with low fertility rates in India. This topic remains relatively underexplored, making the study particularly relevant in the Indian context. After an initial round of revisions, the manuscript is now well-organized and clearly written. The following are some minor suggestions that may further improve clarity:

Reply: Thank you very much for your encouraging feedback and for acknowledging the relevance and clarity of our revised manuscript. We sincerely appreciate your constructive suggestions and have carefully considered each of them to further improve the clarity and overall quality of the paper.

Comment 1. Kindly assess the level of multicollinearity—particularly among variables such as age, age at marriage, and age at first birth—prior to conducting the regression analysis, and include the findings in the manuscript.

Reply: Thank you for your valuable suggestion. We have assessed multicollinearity among the key age-related variables (age, age at marriage, and age at first birth) using the Variance Inflation Factor (VIF). All values were below 5 (VIF range: 1.02–3.54), indicating no major multicollinearity concern. This information has been included in the revised methodology section.

Comment 2. Please examine the potential presence of endogeneity, as certain variables used in constructing the Reproductive Health Index (e.g., history of stillbirth/miscarriage, type of delivery assistance) may influence the number of children ever born (CEB). A rationale for the absence of endogeneity in the analysis should be provided.

Reply: Thank you for this valuable comment. We agree that certain indicators, such as history of miscarriage or type of delivery assistance, may conceptually influence or be influenced by fertility behaviours. To address this, we conducted a sensitivity analysis using alternate specifications of the Reproductive Health Index (RHI), including versions that exclude these potentially endogenous components. The results remained consistent in direction and significance, suggesting that any endogeneity is unlikely to meaningfully bias the findings. As this assessment is already reflected in our supplementary analysis (see S1 Table), we have not added a separate discussion in the manuscript.

Comment 3. Specify the states included in each of the six regional categories, or cite the source from which the regional classification was derived.

Reply: Thank you for this suggestion. We have now specified the states included in each of the six regional categories in the Methodology section of the revised manuscript.

Comment 4. In the methodology section, please justify the use of the Poisson regression model. Also, check for the presence of overdispersion or under dispersion in the data and report the findings.

Reply: Thank you for the suggestion. While the rationale for using the Poisson model was already included in the methodology section, we have now tested for overdispersion using the Pearson chi-square goodness-of-fit test. The dispersion statistic was close to 1 (χ² = 45,143.74, p = 1.000), indicating that the Poisson model is well-suited. This clarification has been added to the revised manuscript.

Comment 5. It is recommended to present Relative Risk Ratios (RRR) in Poisson regression results instead of regression coefficients for clearer interpretation of the results.

Reply: Thank you for this suggestion. We have now updated Table 4 and the associated text to report exponentiated Poisson regression results, presenting Incidence Rate Ratios (IRRs) for clearer interpretation. This change enhances the readability and understanding of the model outputs.

Comment 6. Ensure that all figures have appropriate titles and are properly numbered.

Reply: Thank you for your suggestion. We have addressed this comment by ensuring that all figures are now properly numbered Fig 1 and Fig 2 and consistently cited in the manuscript.

Reviewer #4: 1. The introduction section includes limited literature; the authors need to add more recent studies.

Reply: Thank you for your valuable suggestion. We have now revised the Introduction section to include more literature.

Comment 2. The authors should provide more information about the NFHS data, including the questions asked, which questions were used in this study, and the topics covered by NFHS.

Reply: Thank you for this valuable comment. In response, we have elaborated on the NFHS data structure, the specific women’s module used, and the variables selected for this study. We now provide greater detail on the questions informing each reproductive health indicator. These clarifications have been added to the Data Sources and Study Variables and Their Measurement.

Comment 3. The study variables and their measurement methods require further details.

Reply: Thank you for your thoughtful comment. We would like to note that the detailed definitions, coding schemes, and rationale for all the variables included in the analysis are presented in Table 1 of the manuscript. This table outlines how each component of the Reproductive Health Index (RHI) is derived from the NFHS-5 data, ensuring transparency and replicability. We have also reviewed the surrounding text to ensure it appropriately references and explains the variables where needed.

Comment 4. In Table 2, the authors report that 75% of respondents have ever experienced a stillbirth or miscarriage. This figure appears inaccurate, as the NFHS report indicates only 7.3% for miscarriage and 0.9% for stillbirth. The authors should verify these figures.

Reply: Thank you for this insightful comment. We have carefully reviewed this issue and would like to clarify that our estimate is based on a subsample of currently married women aged 15–49 who had at least one birth in the five years preceding the survey, rather than all women of reproductive age. This differs methodologically from the NFHS-5 national report, which calculates the prevalence of stillbirth and miscarriage across the entire sample of surveyed women. Furthermore, our operationalisation of the variable “ever experienced stillbirth/miscarriage” includes only stillbirth and miscarriage, and excludes abortion, in line with the focus of our Reproductive Health Index (RHI) on involuntary and health-related pregnancy losses. Including abortion, a decision that is often voluntary would not align with the conceptual objective of the index and might misrepresent the reproductive health burden the index is intended to capture. We have clearly noted this in the main text of the Methods section and a footnote has been included below Table 1 to reinforce this decision.

Comment 5. In Table 3, the mean RHI varies by women’s age, age at first marriage, and age at first birth. While RHI increases with higher age at first marriage and first birth, it appears to decrease with advancing age of women. This pattern needs further explanation from the authors to clarify the underlying reasons.

Reply: Thank you for pointing this out. We agree that this is an important pattern that requires clarification. We have now added a brief explanation in the revised results section. While women who marry or give birth at later ages tend to have better reproductive health, likely due to greater physical readiness and increased care-seeking, RHI tends to decline among older women. This could reflect the cumulative effects of reproductive strain, repeated childbearing, or reduced access to health services over time.

---

## [Editor Report · Decision Letter 2]

16 Jul 2025

Does Low Fertility Indicate Better Reproductive Health Status? Evidence from Nationally Representative Survey in India

PONE-D-25-01359R2

Dear Dr. Sikdar,

We’re pleased to inform you that your manuscript has been judged scientifically suitable for publication and will be formally accepted for publication once it meets all outstanding technical requirements.

Kind regards,

Pijush Kanti Khan, Ph.D.

Academic Editor

PLOS ONE

---

## [Editor Report · Acceptance letter]

PONE-D-25-01359R2

PLOS ONE

Dear Dr. Sikdar,

I'm pleased to inform you that your manuscript has been deemed suitable for publication in PLOS ONE. Congratulations! Your manuscript is now being handed over to our production team.

Kind regards,

on behalf of

Dr. Pijush Kanti Khan

Academic Editor

PLOS ONE